# Factors Influencing Self-Reported Medication Use in the Survey of Health Aging and Retirement in Europe (SHARE) Dataset

**DOI:** 10.3390/healthcare9121752

**Published:** 2021-12-18

**Authors:** Aline Schönenberg, Tino Prell

**Affiliations:** Department of Geriatrics, Halle University Hospital, 06120 Halle, Germany; tino.prell@uk-halle.de

**Keywords:** self-report, medication adherence, medication knowledge, depression

## Abstract

The validity of self-reported medication use in epidemiological studies is an important issue in healthcare research. Here we investigated factors influencing self-reported medication use for multiple diagnoses in the seventh wave of the Survey of Health Aging and Retirement in Europe (SHARE) dataset in *n* = 77,261 participants (ages: mean = 68.47, standard deviation = 10.03 years). The influence of mental, physical, and sociodemographic parameters on medication self-report was analyzed with logistic regressions and mediation models. Depression, memory function, and polypharmacy influenced the self-report of medication use in distinct disorders to varying degrees. In addition, sociodemographic factors, knowledge about diagnosis, the presence of several chronic illnesses, and restrictions of daily instrumental activities explained the largest proportion of variance. In the mediation model, polypharmacy had an indirect effect via depression and memory on self-reported medication use. Factors influencing medication self-report vary between different diagnoses, highlighting the complexity of medication knowledge. Therefore, it is essential to assess the individual parameters and their effect on medication behavior. Relying solely on medication self-reports is insufficient, as there is no way to gage their reliability. Thus, self-reported medication intake should be used with caution to indicate the actual medication knowledge and use.

## 1. Introduction

The Survey of Health Aging and Retirement in Europe (SHARE) is a cross-national panel study that includes data on health, social and economic status, and social and family networks of the representative samples of community-based populations from many European countries [1]. So far, several studies have explored the aspects of polypharmacy or the association between distinct medication records and clinical or social factors, e.g., sleep medication use with a sleep disorder or pain medication use with musculoskeletal pain [2,3,4]. However, the primary factors driving the knowledge of medication and behavior remain unidentified [5]. In SHARE, medication use and medical diagnoses are recorded via self-reports for predefined selection of drug classes or disorders, e.g., drugs for hypertension [6,7]. However, this approach has some limitations that must be acknowledged when using self-reported data to proxy for actual medication knowledge or usage. Studies based on self-reported medication use highly depend on the accuracy of those self-reports to draw accurate and reliable conclusions from the available data [8]. However, self-reported medication behavior can be measured using a large bandwidth of different methods [8,9,10], and no consensus has yet been reached regarding its reliability and best practices. For instance, some studies found self-report of medication to be reliable and comparable to objective measures such as pill count, whereas other studies raise the issue that self-report is subjected to recall errors and bias [9,11,12,13]. Errors in self-reporting of medication are linked to general knowledge about medication, attitudes toward health and treatment, and adherence to medication [14]. Particularly, knowledge about the prescribed and self-reported medication is influenced by cognitive function, mood disorders, and the number of drugs taken [12,13,14,15,16]. As an alternative, the accuracy of self-reported medication use can be verified by comparing it with other measures, e.g., prescriptions, healthcare insurance claims, or general practice medical records [10,17,18,19].

However, although available, these objective reports are not frequently used in clinical research and furthermore do not necessarily capture actual medication intake either [19]. Therefore, it is crucial to understand the parameters that influence self-reports of medication [12,13,20]. Several influential factors have previously been proposed, including cognitive deficits (poor recall) or intended nondisclosure, age, polypharmacy, poor health, and social desirability bias or stigmatization [12,13,21,22,23,24,25,26]. However, the influence of these factors varies between studies and diagnoses, and no consensus has been reached regarding their effects [5,14]. Given the high impact of cognitive deficits, mood disorders, physical functioning, and polypharmacy on medication knowledge and behavior [14], we aimed to study the effect of these factors on self-reporting of medication for several diagnoses provided in the SHARE dataset [6].

## 2. Materials and Methods

### 2.1. Study Sample

SHARE is a biennial longitudinal survey of the aging process in individuals in several European Union countries and Israel. The survey collects a multitude of information regarding health, socioeconomic status, and social and family networks. Details about the sampling procedure have been published elsewhere [1]. So far, SHARE has conducted eight panel waves. Here, we analyzed data from the seventh wave (release 7-1-1), which were collected between March and October 2017. SHARE targets all persons above 50 years of age who speak the country’s primary language, who are physically and mentally able to participate, and who are not institutionalized/in hospital or out of the country during the sampling time. Further details about the sampling procedures and criteria can be found in the respective documentation files [1,6,7]. An overview of the study cohort included in the present analyses is provided in Table 1.

### 2.2. Extracted Variables

Data on medication use were collected by asking participants, “Do you currently take drugs at least once a week for (1) high blood cholesterol, (2) high blood pressure, (3) coronary diseases, (4) other heart diseases, (6) diabetes, (7) joint pain, (8) other pain, (9) sleep problems, (10) anxiety or depression, (11) osteoporosis, (12) stomach burns, (13) chronic bronchitis, (14) suppressing inflammation (only glucocorticoids or steroids), (15) none, (16) other?” Participants had the option to choose “refuse,” “don’t know,” “select,” or “not select.”

In addition, the following variables were extracted from the dataset:

The presence of distinct disorders “ever diagnosed/currently having”: (1) heart attack, (2) high blood pressure or hypertension, (3) high blood cholesterol, (4) stroke, (5) diabetes or high blood sugar, (6) chronic lung disease, (7) cancer, (8) stomach or duodenal ulcer, peptic ulcer, (9) Parkinson’s disease, (10) cataracts, (11) hip fracture or femoral fracture, (12) other fractures, (13) Alzheimer’s disease, dementia, senility, (14) other affective/emotional disorders, (15) rheumatoid arthritis, (16) osteoarthritis/other rheumatism, (17) chronic kidney disease, (18) none, (19) other. Participants had the option to choose “refuse,” “don’t know,” “select,” or “not select.” The variable “Number of chronic diseases” was based on the number of chronic diseases reported by each individual.

Regarding sociodemographic factors, age was calculated according to “Year of birth” subtracted from 2017, when wave 7 was conducted. Gender had “male” or “female” as possible answers.

In addition to sociodemographic factors, based on the previously cited literature, cognition, restrictions in performance of daily activities, and mood were included as further covariates to understand how self-report of medication can be influenced.

To estimate patients’ cognition, the 10-word recall test was used for immediate and delayed episodic memory [27,28]. The test consisted of verbal registration and recall of a list of ten words immediately (first trial) and once after a delay time (delayed recall). The total scores of the two tests ranged from 0 to 10 and corresponded to the number of words the respondent could recall. Baseline cognitive function in SHARE was also measured in verbal fluency and numeracy. Based on previous studies, forgetting to take medication was identified as the most common reason for nonadherent behavior [25]; therefore, we used delayed episodic memory (delayed recall) as a cognitive measure in our study. Patients were categorized into poorer memory (<5 words correct) and better memory (≥5 words correct) groups [6,7].

The limitations with instrumental activities of everyday life (IADL) index [29] was used to describe the number of limitations with seven instrumental activities of everyday life [30]. It ranges from 0 to 7, with higher values indicating more difficulties with these activities and thus impaired mobility of the respondent.

Depression was defined using the total score on the EURO-D scale [31]. It covers 12 symptom domains: depressed mood, pessimism, suicidality, guilt, sleep, interest, irritability, appetite, fatigue, concentration, enjoyment, and tearfulness. Each item is scored 0 (symptom not present) or 1 (symptom present), and the item scores are summed to produce a scale ranging from 0 to 12. Respondents were divided into non-depressed (EURO-D <3) and depressed (EURO-D ≥4) groups [6,7,31,32,33].

### 2.3. Statistical Analysis

All statistical analyses were performed with R version 3.6.2 (R Foundation for Statistical Computing, Vienna, Austria), with a *p*-value < 0.05 indicating statistical significance. Values were given as means and standard deviations, and categorical variables were presented as numbers or percentages. Descriptive statistics were used to characterize the cohort. The association between several parameters and self-reported medication use was studied using univariate Spearman’s correlation and binomial logistic regression models (backward selection). The significance levels for variables entering the linear regression model and removing from the model were set at 0.05 and 0.1, respectively. We excluded autocorrelation and multicollinearity prior to analysis (variance inflation factor and tolerance) [34].

A mediation model was used to study the impact of polypharmacy on self-reported medication use [35,36]. Memory and depression were used as moderator(s) of the relationship between polypharmacy and self-reported medication use. Additionally, age, sex, IADL, and the number of chronic disorders were entered as covariates. The statistical significance of the direct and indirect effects was evaluated using 10,000 bootstrap samples to create bias-corrected confidence intervals (CIs; 95%). The relationship between all variables involved in the mediation analysis was approximately linear, as assessed by visually inspecting the scatterplots after LOESS smoothing. Since the pure effect of mediation is described by the indirect effect, this is the most important criterion for mediation, regardless of the other prerequisites [37,38].

## 3. Results

The descriptive statistics of the investigated cohort are provided in Table 1 and detailed in the initial dataset publications [6].

The most common self-reported disorders and conditions were hypertension, high blood cholesterol, and osteoarthritis (Figure 1A). Accordingly, most self-reported drugs were prescribed for these conditions (Figure 1B).

In the univariate analyses, the frequency and ratio between selected and not-selected drugs differed between people with poor and better memory (Figure 2A) and between those with and without depression (Figure 2B).

The reports of distinct disorders and—if available—the use of corresponding drugs were highly correlated, indicating that people who can select the correct diagnosis can also select the appropriate medication (Table 2). These correlations were comparable between patients with good and poor memory function (Appendix A Appendix A).

Logistic regression was used to further study the association between selected/not selected distinct drug classes, with memory (delayed recall of ten-word list), depression (EURO-D), gender, age, the number of chronic disorders, polypharmacy (yes/no), and selected corresponding diagnosis (present/absent) as independent variables. Correlations between predictor variables were low (*r* < 0.70), indicating that multicollinearity was not a confounding factor in the analysis. Exemplary findings on hypertension are presented in the paragraphs below; further analyses on other diagnoses are given in the Appendix A.

As one example, the explained variance of memory, depression, and polypharmacy was 6% for antihypertensive drugs (Nagelkerke’s *R*² = 0.057) (Model 1 in Table 3). After entering age, the number of chronic disorders, IADL, gender, and report of hypertension as a diagnoses as additional independent variables, the explained variance increased to 72%. However, in the final model, memory and depression were no longer the significant predictors of the self-reported use of antihypertensive drugs (Model 2 in Table 3). Findings for drugs used for diabetes, chronic lung diseases, high cholesterol, anxiety, pain, and stomach problems are provided in Appendix A. Results for drugs where no single corresponding disorder was recorded are provided in Appendix A.

Of note, 1885 (2.5%) participants reported having dementia in the list of available diagnoses. There was a weak correlation between delayed recall of the ten-word list and the presence of dementia (*r* = 22,120.16, *p* < 0.001). As this might cause bias, analyses were repeated after the exclusion of people with dementia. This did not remarkably change the former results (Appendix A Appendix A).

In summary, the combination of polypharmacy, memory deficits, and depression contributed in varying degrees to the self-report of medication use (Figure 3).

As demonstrated in the logistic regressions for all diagnoses assessed in this analysis, polypharmacy influences self-reported medication. Therefore, we used a mediation model to explore how polypharmacy exerts this influence. Based on previous literature, we hypothesized that the effect of polypharmacy could be mediated by depression (i.e., nondisclosure) or cognitive deficits (i.e., memory problems).

A simple mediation model is displayed in Figure 4A for the self-report of antihypertensive medication use. In this initial model, polypharmacy had a significant direct effect on the self-reported medication. In addition, there was an indirect effect of polypharmacy via depression and memory on self-reported medication (Figure 4). After entering covariates to the model, only the indirect effect through depression remained significant (Figure 4B). For diabetic medication as another example of a frequent diagnosis, both depression and memory mediated the effect of polypharmacy on self-reported diabetes medication use (Figure 4C) (see Appendix A Appendix A for the full model).

## 4. Discussion

Using the SHARE dataset, which contains representative data on many diagnoses, we were able to demonstrate that various factors, including polypharmacy, several chronic disorders, knowledge about the diagnosis, depression, memory, restrictions in IADL, and gender might influence the self-reported medication use to different degrees. As in previous studies, all the factors, except for polypharmacy and knowledge of diagnosis, varied in their influence on the diagnoses [12,14,21].

As shown in previous studies, polypharmacy influences the self-report of medication. An increasing number of daily medications is linked to poorer knowledge when medication regimes become too complex [12,25,39]. Polypharmacy is also associated with increased age, depression, memory loss, and poor physical functioning [40] and it can be regarded as an indicator of general poorer health [14,41]. Therefore, it is possible that patients taking various medications are more likely to forget some of them when asked to report their medication. Similarly, as shown in our mediation analysis, polypharmacy itself was linked to depression, memory, and sociodemographic factors. This relationship also shows that factors related to medication behavior are not fully separable. Furthermore, they may be linked to the report itself and to the probability of getting specific prescriptions because of health reasons. The influence of polypharmacy in particular is highly relevant, as polypharmacy poses a risk for patients since not all prescribed medications are necessarily appropriate. Several methods have been developed to detect such inappropriate medication; however, it is essential to keep in mind the influence of the number of medications alone on self-reported medication use [42,43,44], to detect potentially harmful medication regimes.

Knowledge about the corresponding diagnoses strongly increased the chances of selecting medications, indicating that both have the same underlying principle of health knowledge. This is also shown by the correlations between the selection of diagnosis and corresponding medication in our analyses [5,12,14].

Similar to polypharmacy, the presence of multiple chronic disorders influenced medication reports [14,21]. However, the direction of this influence varied across diagnoses. Patients who might be accustomed to having an illness for a long time might remember them better. However, multiple chronic disorders might be linked to overall poor health and polypharmacy, again leading to difficulties in reporting all medications [14,45].

Depression and cognition have often been recognized as critical factors influencing medication behavior [5,14,15], although their influence varied across the different diagnoses reported in this study. It is essential to keep in mind that like polypharmacy, depression and cognition are reciprocally linked to health and may affect the medication report itself and actual prescription probability as health factors. Therefore, it is crucial to understand how depression and cognition influence the medication report itself, e.g., via motivational deficits, and how they are related to medication behavior and actual health. Our analyses revealed no difference in the results when excluding people reporting dementia and showed that cognition had little effect on knowledge of diagnosis and medication. However, it should be noted that there was no neuropsychological testing involved in the SHARE data collection, but the proportion of participants with reported cognitive disorders was low [1,6,7,46]. This comparability of results across people with and without cognitive impairment suggests that although cognition may influence medication reports, there is a difference between the general report of the overall presence of medication and more finely tuned knowledge, such as the exact name, dosage, or time of administration [14,25,47]. The varying direction of influence of depression and cognition in this study suggests that they are not exclusively linked to the medication report as would be expected if they had exerted their influence through nondisclosure due to the lack of motivation or forgetting. However, as shown in the mediation models, depression, polypharmacy, and cognition may not only be driving forces influencing medication knowledge but also generally concomitant symptoms of specific diagnoses and associated health factors.

Accordingly, depression, memory, and polypharmacy did not explain much of the variance of medication selection on their own. This indicates that other factors must be considered, such as age, number of diagnoses, gender differences, IADL, and the knowledge of the diagnosis. However, these factors also varied between different diagnoses, again highlighting the complexity and individuality of medication knowledge [5,12,45].

One necessary restriction of self-reported medication is the dissociation between medication knowledge, i.e., reporting prescribed medication, and actual medication behavior (e.g., taking it, adherence). This analysis only assessed self-reported medication knowledge on a superficial level. The often-cited impact of depression and cognition may act on actual medication behavior and adherence more than on the report alone. However, there was no objective measure of medication in this dataset, such as counting pills or comparison with pharmacy records [5,13]. Therefore, an objective assessment of actual medication for comparison is not provided; statements can only be made about the overall knowledge about prescribed medication and not about the medication behavior. Similarly, no assessment can be made for patients who did not select medications because there is no objective measure assessing if this information is correct. This again highlights the importance of including objective measures of medication taken to gage the adherence levels of patients.

Although the SHARE dataset presents several advantages, especially the large sample size and the inclusion of 28 different countries, it also poses certain limitations. First, SHARE is based on self-report that may not capture the medication knowledge or health status of the participants accurately [9,10]. Similarly, people who repeatedly agree to participate in large-scale research projects are more likely to be healthy and motivated. Thus, it is possible that a large group of people with more severe health problems is not represented in this survey, mainly as only noninstitutionalized participants were recruited. In SHARE, the magnitude of nonresponse and panel attrition may generate sample selection bias, limiting the representativeness of the database and the generalizability of results [7].

Similarly, reports across several countries with varying health systems and living standards may not be entirely comparable, despite the advantages promised by internationality and large sample sizes. However, the SHARE data are much more complex than conventional survey data due to their cross-national and multidisciplinary nature. This enables the exploration of the complex relationship between different life domains, individuals, family, social networks, states, and across the entire life course over time [6,7]. Furthermore, for this study, neither depression nor cognition was assessed in clinical testing. However, both measures have been validated previously and are frequently used in the literature [28,31,33].

Overall, although self-reports have been reported as relatively comparable to objective measures that can be helpful to find individual problems with medication [10,11], the results of this analysis reveal that medication self-reports are influenced by various factors, suggesting that they do not necessarily portray actual medication behavior [5,13,45]. Factors influencing medication reports vary between diagnoses, highlighting the complexity of medication knowledge and adherence and the need for assessing the interplay of parameters and their effect on medication behavior individually [12,13,23,45,47,48]. Relying solely on medication self-reports is insufficient, as there is no way to gage the validity of those reports. Based on the self-report alone, it is impossible to tell if the identified variables of influence affect the self-report alone, the actual health status, or both. Thus, in scientific research and clinical practice, medication self-reports should be used with caution as single indicators of medication behavior.

## Figures and Tables

**Figure 1 healthcare-09-01752-f001:**
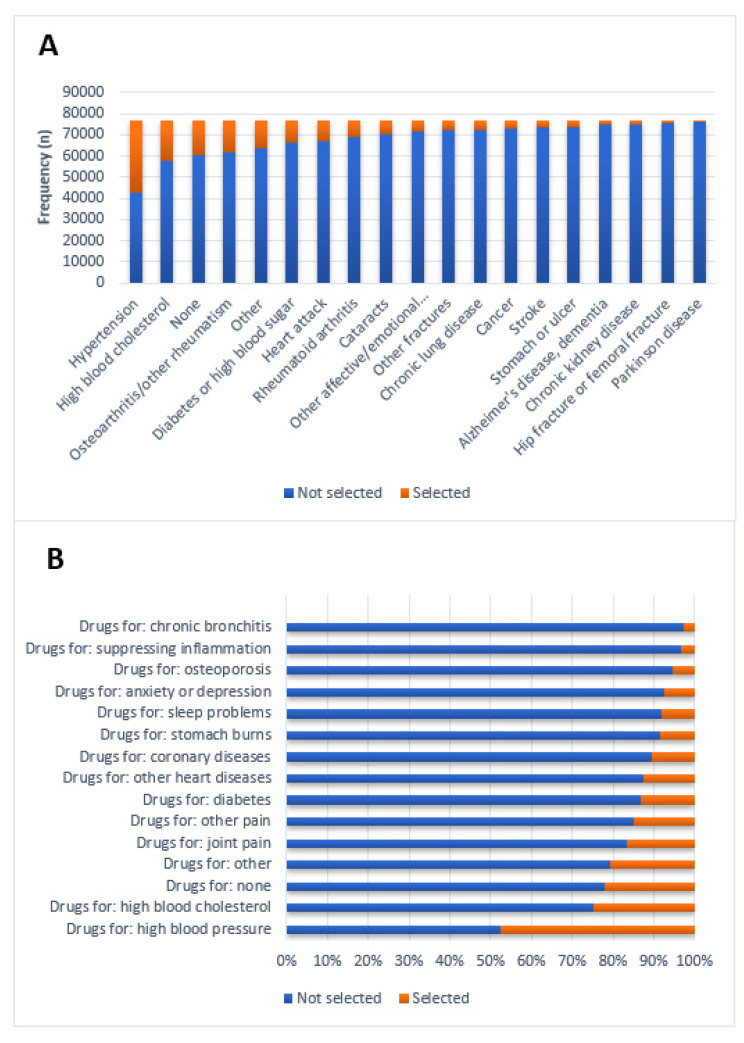
(**A**) Frequencies of self-reported diseases and (**B**) frequencies of self-reported medication use (selected) or nonuse (not selected) in the entire cohort (*n* = 76,876).

**Figure 2 healthcare-09-01752-f002:**
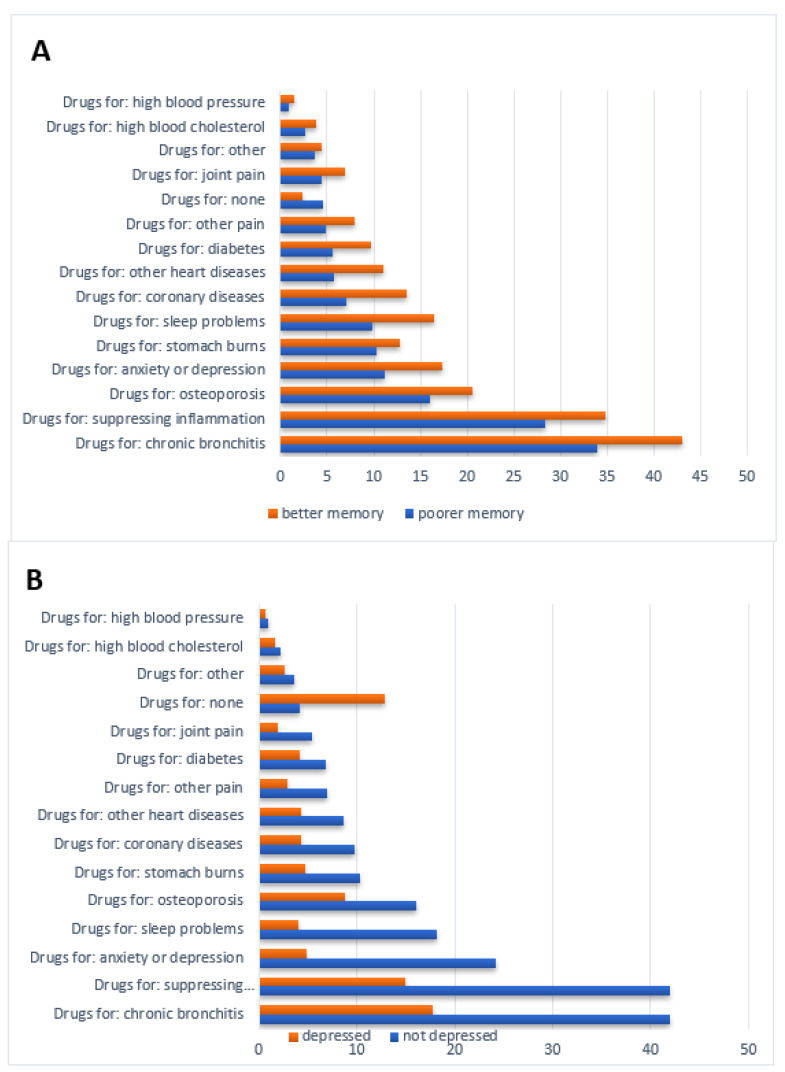
(**A**) The ratio of drugs not selected/selected for use depending on memory function as indicated by the ten-word list learning delayed recall (poorer memory <5 words correct, better memory ≥5 words correct). (**B**) The ratio of drugs not selected/selected for use depending on their depression status (EURO-D <3 or ≥4).

**Figure 3 healthcare-09-01752-f003:**
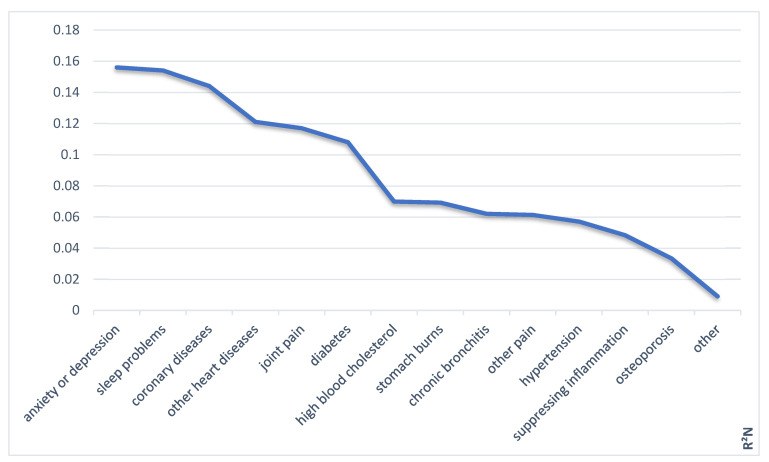
Explained variance (R²_N_) of self-reported medication use by memory, depression, and polypharmacy in corresponding binominal logistic regressions.

**Figure 4 healthcare-09-01752-f004:**
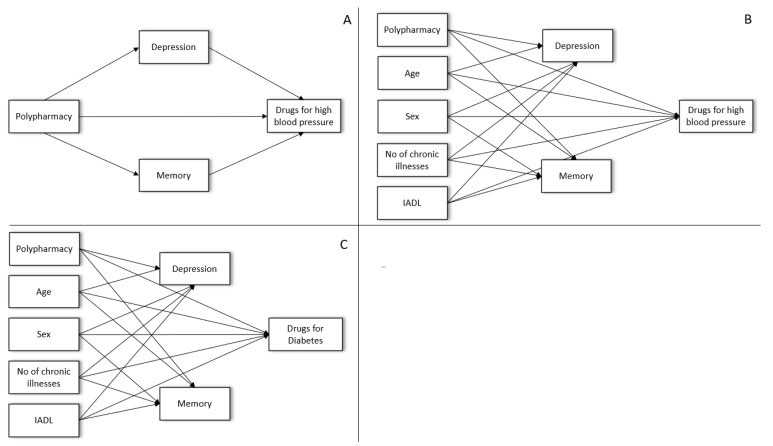
Mediation models. (**A**) Simple mediation model for hypertension, including only depression and memory. (**B**) A comprehensive mediation model for hypertension. (**C**). Comprehensive mediation model for diabetes. Note: IADL = instrumental activities of daily living.

**Table 1 healthcare-09-01752-t001:** Summary of cohort characteristics.

Variable		M	Mdn	SD	IQR
Age (years)		68.5	68	10.0	61–76
Depression (EURO-D)		2.4	2	2.3	1–4
Memory (delayed recall for ten words list learning)		3.7	4	2.2	2–5
		*n*		%	
Sex	Male	33,150		42.9	
	Female	44,111		57.1	
Number of chronic diseases	Refusal	39		0.1	
	Don’t know	118		0.2	
	0	16,061		20.9	
	1	21,187		27.6	
	2	16,463		21.4	
	3	11,096		14.4	
	4	6233		8.1	
	5	3123		4.1	
	6	1455		1.9	
	7	627		0.8	
	8	295		0.4	
	9	116		0.2	
	10	42		0.1	
	11	18		0.0	
	12	7		0.0	
	13	2		0.0	
Limitations with instrumental activities of daily living	Refusal	35		0.0	
	Don’t know	107		0.1	
	0	61,575		80.6	
	1	5919		7.8	
	2	2367		3.1	
	3	1592		2.1	
	4	1158		1.5	
	5	827		1.1	
	6	679		0.9	
	7	633		0.8	
	8	487		0.6	
	9	980		1.3	
Taking at least five different drugs on a typical day	Refusal	20		0.0	
	Don’t know	135		0.2	
	Yes	18,320		30.5	
	No	41,615		69.3	

Note: M = mean, Mdr = median, IQR = interquartile range, SD = standard deviation.

**Table 2 healthcare-09-01752-t002:** Correlation between the reporting of distinct disorders and drug use.

Ever Diagnosed/Currently Have	Drugs for													
	High Blood Cholesterol	High Blood Pressure	Coronary Diseases	Other Heart Diseases	Diabetes	Chronic Bronchitis	Anxiety or Depression	Joint Pain	Stomach Burns	Other Pain	Sleep Problems	Osteoporosis	Suppressing Inflammation	None
High blood cholesterol	**0.704**	0.243	0.145	0.121	0.170	0.052	0.074	0.094	0.119	0.067	0.095	0.073	0.037	−0.229
High blood pressure or hypertension	0.255	**0.825**	0.153	0.163	0.190	0.053	0.056	0.128	0.088	0.084	0.083	0.049	0.041	−0.421
Heart attack	0.182	0.174	**0.437**	**0.513**	0.107	0.076	0.052	0.098	0.087	0.073	0.096	0.042	0.053	−0.174
Diabetes or high blood sugar	0.221	0.213	0.128	0.101	**0.882**	0.052	0.057	0.088	0.072	0.056	0.073	0.022	0.035	−0.189
Chronic lung disease	0.052	0.063	0.097	0.100	0.052	**0.524**	0.070	0.103	0.102	0.080	0.087	0.074	0.096	−0.082
Other affective/ emotional disorders	0.057	0.045	0.062	0.066	0.041	0.078	**0.537**	0.119	0.122	0.133	0.257	0.081	0.074	−0.098
Rheumatoid arthritis	0.069	0.107	0.073	0.093	0.069	0.084	0.093	**0.359**	0.113	0.171	0.114	0.155	0.134	−0.123
Osteoarthritis/other rheumatism	0.081	0.088	0.063	0.072	0.041	0.076	0.088	**0.338**	0.141	0.173	0.123	0.224	0.114	−0.137
Stomach or duodenal ulcer, peptic ulcer	0.059	0.057	0.072	0.074	0.040	0.090	0.089	0.111	**0.330**	0.116	0.098	0.084	0.078	−0.064
Alzheimer’s disease, dementia, senility	0.037	0.036	0.088	0.055	0.051	0.047	0.146	0.060	0.043	0.069	0.114	0.049	0.045	−0.057
Stroke	0.109	0.128	0.313	0.150	0.084	0.066	0.081	0.070	0.068	0.077	0.098	0.050	0.051	−0.087

Corresponding drug use and disorders are highlighted in bold. *n* = 76,743, for all correlations *p* < 0.001.

**Table 3 healthcare-09-01752-t003:** Exemplary binomial logistic regression: drugs for hypertension.

Predictor.	*p*	Odds Ratio	95% CI Lower	95% CI Upper	Model Fit Measures		Overall Model Test		
					R²_N_	AIC	χ²	df	*p*
Model 1									
Intercept	<0.001	4.162	3.673	4.715	0.0570	14330	477	3	< 0.001
Depression	<0.001	0.956	0.939	0.973					
Memory	0.007	0.974	0.956	0.993					
Polypharmacy No–Yes	<0.001	0.372	0.338	0.409					
Model 2									
Intercept	<0.001	0.0413	0.0206	0.0825	0.7221	6379	8439	8	<0.001
Depression	0.333	0.9846	0.9540	1.0160					
Memory	0.395	1.0146	0.9813	1.0491					
Polypharmacy No–Yes	<0.001	0.3342	0.2849	0.3920					
Age	<0.001	1.0403	1.0316	1.0491					
Number of chronic Illnesses	<0.001	0.8597	0.8137	0.9084					
Instrumental activities of daily life (IADL)	0.022	0.9474	0.9047	0.9922					
Sex: Female–Male	0.031	0.8637	0.7559	0.9868					
Hypertension: Selected–Not selected	<0.001	156.4674	130.6701	187.3576					

## Data Availability

This study uses data from SHARE Wave 7 (10.6103/SHARE.w7.711), see Börsch-Supan et al. (2013) for methodological details [1]. Further information on the seventh wave can be found at Bergmann et al. (2019) and Börsch-Supan et al. (2020) [6,7]. Data collected and generated in the SHARE projects are available free of charge for scientific research without any restrictions on specific research purposes. The SHARE data collection has been funded by the European Commission through FP5 (QLK6-CT-2001-00360), FP6 (SHARE-I3: RII-CT-2006-062193, COMPARE: CIT5-CT-2005-028857, SHARELIFE: CIT4-CT-2006-028812), FP7 (SHARE-PREP: GA N°211909, SHARE-LEAP: GA N°227822, SHARE M4: GA N°261982, DASISH: GA N°283646) and Horizon 2020 (SHARE-DEV3: GA N°676536, SHARE-COHsESION: GA N°870628, SERISS: GA N°654221, SSHOC: GA N°823782) and by DG Employment, Social Affairs and Inclusion. Additional funding from the German Ministry of Education and Research, the Max Planck Society for the Advancement of Science, the U.S. National Institute on Aging (U01_AG09740-13S2, P01_AG005842, P01_AG08291, P30_AG12815, R21_AG025169, Y1-AG-4553-01, IAG_BSR06-11, OGHA_04-064, HHSN271201300071C) and from various national funding sources is gratefully acknowledged (see www.share-project.org, last accessed on 23 November 2021).

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
