# Peer review of "Factors Influencing Self-Reported Medication Use in the Survey of Health Aging and Retirement in Europe (SHARE) Dataset"

_healthcare, 2021, doi:10.3390/healthcare9121752_

Round 1

Reviewer 1 Report

Dear Author,

This manuscript is a well-prepared article identifying the factors influencing self-reported medication use for multiple diagnoses in the seventh wave of the Survey of Health Ageing and Retirement in Europe (SHARE) dataset. Accordingly, depression, cognition, and polypharmacy influenced the self-report of medication use in distinct disorders to varying degrees. However, I have some suggestions that will contribute to the article scientifically.

1-         For the whole article, English and spelling rules should be revised and improved.

2-         The abstract should be about 200 words maximum and should follow the style of structured abstracts but without headings.

3-         It would be better to state the date the study was conducted, how long the data were collected, and the inclusion/exclusion criteria in the ‘Materials and methods' section.

4-         Didn't you collect data on whether participants took medication for stroke, cancer, Parkinson's disease, cataracts, dementia, and/or chronic kidney disease?

5-        Why are the activities of daily living (ADL) not included as a covariate while having the IADL?

5-        How did you choose the covariates? Did you choose them according to the factors associated with the self-reported medication use in the literature?

6-         In order to give a global view and discuss, it would be better to compare polypharmacy and inappropriate medication use percentages by giving data from Turkey and other countries.

7-         It would be better to mention briefly about common and most updated inappropriate medication use tools (STOPP-START, Beers, Time criteria etc.) where appropriate.

8-         PMID: 34653922 can also be mentioned.

9-         The references older than ten years should be updated, if possible.

Author Response

Dear Editor, Dear Reviewer,

We are grateful for the detailed feedback on our manuscript and the opportunity to resubmit an improved version of our work. In the revised version, we have incorporated the mentioned points to the best of our abilities and all changes made are tracked in the manuscript. Please find below the requested response to the Reviewer's comments.

COMMENT:

1- For the whole article, English and spelling rules should be revised and improved.

RESPONSE:

We thank the Reviewer for this comment. Prior to submission, the article was reviewed by an experienced editor whose first language is English and who specialises in editing scientific manuscripts. In addition, the article has been re-read and corrected by a colleague who speaks English on native speaker level. We hope that the article is now up to the standards Healthcare requires for publication.

COMMENT:

2- The abstract should be about 200 words maximum and should follow the style of structured abstracts but without headings.

RESPONSE:

We kindly thank the reviewer for pointing out this oversight on our part; we have removed the headings, taking the abstract to less than 200 words.

COMMENT:

3- It would be better to state the date the study was conducted, how long the data were collected, and the inclusion/exclusion criteria in the ‘Materials and methods' section.

RESPONSE:

As SHARE is a Europe-wide data collection project, all relevant information concerning the project is well-documented and can be found in the cited references. However, for ease of understanding of our manuscript, we have added more information on the data collection procedure to the respective section.

COMMENT:

4- Didn't you collect data on whether participants took medication for stroke, cancer, Parkinson's disease, cataracts, dementia, and/or chronic kidney disease?

RESPONSE:

We thank the Reviewer for this question. Yes, information on whether those diagnoses are present was collected in the dataset. However, the main goal of our analysis was to show how easily self-report of medication can be influenced and how it varies according to a multitude of factors. Since the presence of these diagnoses is based on self-report and no doctor consultation was performed, making a statement about particular diagnoses and their relationship with self-reported medication should be done only with caution. Instead, we chose a multitude of the most common diagnoses for the sole purpose of showing how the disease itself can influence self-report. However, it is neither the intention nor within the scope of this analysis to present information on every available diagnosis. If requested, logistic regressions can be added for additional diagnoses, but we do not consider it useful to include analyses on such heterogeneous illnesses as cancer and stroke, especially when no further information is available on the illness, or on dementia. Furthermore, the number of participants with Parkinson’s Disease was comparatively low (less than 700 for PD and more than 14.000 for osteoarthritis/rheumatism).

COMMENT:

5- Why are the activities of daily living (ADL) not included as a covariate while having the IADL? How did you choose the covariates? Did you choose them according to the factors associated with the self-reported medication use in the literature?

RESPONSE:

We thank the Reviewer for these questions. As the participants in the SHARE dataset were exclusively non-institutionalized persons who were able to participate in a complex survey with several assessments, it is probable that the participants in SHARE are overall comparably healthy. Therefore, we decided to use IADL instead of ADL as a measure of restrictions in daily life; we assumed this to be more appropriate since restrictions in IADL are more likely to be already present in persons with relatively high functional status.

The covariates were indeed chosen based on previous research, as indicated in the introduction, especially the final paragraph. To clarify this procedure, we have included a statement about the selection of covariates in the methods section.

COMMENT:

  1. In order to give a global view and discuss, it would be better to compare polypharmacy and inappropriate medication use percentages by giving data from Turkey and other countries.

RESPONSE: We thank the Reviewer for this suggestion. We did not include country-specific information in our analysis, because the aim of our analysis was not to provide a country-specific overview. Rather, it was to show how many factors other than the medication itself can influence the self-report of those medications in a large cohort, and thus to raise the level of caution when using self-reports as an indicator of actual medication knowledge and intake. Based on the data we have included in our analysis, it is not possible to draw conclusions on the specific situation in a particular country, as a large amount of country-specific information would have to be included to make those comparisons feasible and to put them into perspective. We fear that simply stating percentages would not do justice to this complex topic and would introduce misunderstandings. While we agree that country-specific information is highly valuable and should be looked at more closely, this was not our intention with the present analysis. Additionally, the data is freely available for scientific use; thus, any researcher interested in country-specific differences can look at those in detail, and the study population has been described in the references cited in the manuscript. Unfortunately, Turkey was not part of the countries in which the SHARE data collection was performed. 

COMMENT:
7- It would be better to mention briefly about common and most updated inappropriate medication use tools (STOPP-START, Beers, Time criteria etc.) where appropriate.

RESPONSE: We are grateful to the Reviewer for raising this important topic. Although inappropriate medication use is not the main topic of our analysis, which is self-report of medication, we agree that the topic is closely linked to polymedication and have thus included some references pertaining to the tools mentioned.

COMMENT:

8- PMID: 34653922 can also be mentioned.

RESPONSE: We thank the Reviewer for providing us with this source; we added it to the manuscript.

COMMENT:

9- The references older than ten years should be updated, if possible.

RESPONSE: We appreciate this comment, however, the vast majority of our references dates back no further than 2010, and we have included a multitude of references from the last 3-4 years. Additionally, some of the older references cite questionnaires or diagnostic procedures such as the EURO-D or the 10 word list for cognitive performance, thus they must be included in the manuscript. If a study is well-conducted, the mere fact that it is a few years old does not automatically make it redundant, and we believe it to be important to provide a mixture of older and newer research to cover all important aspects. Additionally, we have paid attention to combine older references and new reference for the respective topic, see for example references 9, 11-13, which are cited together and refer to the years 1999, 2000, 2013 and 2015. However, in those cases where only older studies were cited, we have added newer results to compliment the older ones. Additionally, we made sure to include only newer references in the citations added to the manuscript during revision.

Reviewer 2 Report

In this manuscript are presented results from an investigation on the factors influencing self-reported medication use in the Survey of Health Ageing and Retirement in Europe (SHARE) dataset. The topic is relevant and the study design and the study procedure are very clear. However, for clinical practice, it is more important the adherence to the treatments. Despite the authors referred to this issue as a limitation of the study, in my opinion, it is more important to study the adherence to the medication than the factors that influence the self-reported medication.

I would like to make some suggestions for revision:

Introduction

Line 33-“ In SHARE, medication use and medical diagnoses are recorded via self-reports for predefined selection of drug classes or disorders, e.g., drugs for  hypertension..” Please provide a reference at the end of each phrase.

Line 38-“ However, self-reported medication behavior is often measured using a large bandwidth of different methods” Please provide a reference at the end of each phrase.

Results

Table 1, All the abbreviations should be speeled. The authors can introduce the spelling of each abbreviation as a footnote in the table. Not all readers know the meaning of M and SD, for example. For the data related to depression and memory what was the median (and the IQR)? Please confirm (according to the data) if the median represents better the data? Authors also must standardize the decimal places used.

Figure 1 and 2- The authors must place the letters A and B in the respective graphics to avoid misunderstandings

Author Response

Dear Editor, Dear Reviewer,

We are grateful for the detailed feedback on our manuscript and the opportunity to resubmit an improved version of our work. In the revised version, we have incorporated the mentioned points to the best of our abilities and all changes made are tracked in the manuscript. Please find below the requested response to the Reviewer's comments.

COMMENT:

Line 33-“ In SHARE, medication use and medical diagnoses are recorded via self-reports for predefined selection of drug classes or disorders, e.g., drugs for  hypertension..” Please provide a reference at the end of each phrase.

Line 38-“ However, self-reported medication behavior is often measured using a large bandwidth of different methods” Please provide a reference at the end of each phrase.

RESPONSE: 

We kindly thank the Reviewer for pointing out these oversights on our part; we have added the respective references to the manuscript.

COMMENT:

Table 1, All the abbreviations should be speeled. The authors can introduce the spelling of each abbreviation as a footnote in the table. Not all readers know the meaning of M and SD, for example. For the data related to depression and memory what was the median (and the IQR)? Please confirm (according to the data) if the median represents better the data? Authors also must standardize the decimal places used.

RESPONSE:

We thank the Reviewer for this comment and have updated the manuscript accordingly. The Median an IQR were added for numerical values.

COMMENT:

Figure 1 and 2- The authors must place the letters A and B in the respective graphics to avoid misunderstandings

RESPONSE:

We are grateful to the Reviewer for drawing our eye to this omission on our part; we updated the figures to provide a better description.

Round 2

Reviewer 2 Report

The authors clearly improved the manuscript, therefore, in my opinion, it is now susceptible for publication